# A potential source of atmospheric sulfate from $O_2^-$-induced $SO_2$ oxidation by ozone

Narcisse Tchinda Tsona and Lin Du

Environment Research Institute, Shandong University, Binhai Road 72, 266237 Shandong, China

*Correspondence to*: Lin Du (lindu@sdu.edu.cn)

**Abstract.** It was formerly demonstrated that $O_2SOO^-$ forms at collisions rate in the gas-phase as a result of $SO_2$ reaction with $O_2^-$. Hereby, we present a theoretical investigation of the chemical fate of $O_2SOO^-$ by reaction with $O_3$ in the gas-phase, based on *ab initio* calculations. Two main mechanisms were found for the title reaction, with fundamentally different products: (i) formation of a van der Waals complex followed by electron transfer and further decomposition to $O_2 + SO_2 + O_3^-$ and (ii)

formation of a molecular complex from $O_2$ switching by $O_3$, followed by $SO_2$ oxidation to $SO_3^-$ within the complex. Both reactions are exergonic, but separated by relatively low energy barriers. The products in the former mechanism would likely initiate other $SO_2$ oxidations as shown in previous studies, whereas the latter mechanism closes a path wherein $SO_2$ is oxidized to $SO_3^-$. The latter reaction is atmospherically relevant since it forms the $SO_3^-$ ion, hereby closing the $SO_2$ oxidation path initiated by $O_2^-$. The main atmospheric fate of $SO_3^-$ is nothing but sulfate formation. Exploration of the reactions kinetics

indicates that the path of reaction (ii) is highly facilitated by humidity. For this path, we found an overall rate constant of $4.0 \times 10^{-11}$ $cm^3$ molecule$^{-1}$ s$^{-1}$ at 298 K and 50% relative humidity. The title reaction provides a new mechanism for sulfate formation from ion-induced $SO_2$ oxidation in the gas-phase and highlights the importance of including such mechanism in modeling sulfate-based aerosol formation rates.

## 1 Introduction

The chemistry of sulfur is highly important in the atmosphere. Through its oxidation products, sulfur participates in the formation of secondary atmospheric aerosols, clouds and acid rain. Sulfur dioxide ($SO_2$), the most abundant sulfur-containing molecule in the atmosphere, is known to react both in the gas-phase and in multiphase oxidation processes following different mechanisms to form sulfate as the final oxidation species. The main $SO_2$ oxidizers in the gas-phase include the hydroxyl radical (OH) (Seinfeld and Pandis, 2016), stabilized Criegee intermediates (Welz et al., 2012; Mauldin III et al., 2012; Vereecken et al., 2012) and atmospheric ions (Fehsenfeld and Ferguson, 1974; Enghoff et al., 2012; Tsona et al., 2015). The main routes for $SO_2$ heterogeneous/multiphase oxidation include reactions with mineral dust (Harris et al., 2013), $O_3$ and $H_2O_2$ in cloud droplets (Caffrey et al., 2001; Hoyle et al., 2016; Harris et al., 2012; Hegg et al., 1996), $NO_2$ and $O_2$ in aerosol water and on $CaCO_3$ particles (Cheng et al., 2016; Wang et al., 2016; Zhang et al., 2018; Yu et al., 2018; Zhao et al., 2018). In the gas-phase, the $SO_2$ oxidation by OH and Criegee intermediates leads to $SO_3$ that ultimately forms $H_2SO_4$ (Larson et al., 2000), whereas reactions with ions are generally more complex. In the aqueous phase, $SO_4^{2-}$ is formed as the terminal oxidation species. Sulfate is known to be the main driving species in atmospheric aerosols formation and its formation is critical in the determination of aerosol formation rates (Nieminen et al., 2009; Sipilä et al., 2010; Kuang et al., 2008; Kulmala et al., 2000). The role of ions in this formation has been well established (Yu, 2006; Yu and Turco, 2000, 2001; Enghoff and Svensmark, 2008; Kirkby et al., 2011; Wagner et al., 2017; Yan et al., 2018), although relatively minor compared to the mechanism involving neutral particles, exclusively (Eisele et al., 2006; Manninen et al., 2010; Kirkby et al., 2011; Hirsikko et al., 2011; Wagner et al., 2017).

The immediate products of $SO_2$ oxidation by ions are mostly sulfur oxides ions intermediates (Fehsenfeld and Ferguson, 1974; Möhler et al., 1992; Bork et al., 2012; Tsona et al., 2014) that are susceptible of triggering new reactions or recombining with oppositely charged counterparts to form neutral species. Some of these ions, namely $SO_3^-$, $SO_4^-$, and $SO_5^-$, were detected at relatively high concentrations in the ambient atmosphere (Ehn et al., 2010) and in chamber experiments of $SO_2$ ionic oxidation studies (Nagato et al., 2005; Hvelplund et al., 2013; Kirkby et al., 2011; Kirkby et al., 2016). The chemical fate of most sulfur oxides anions is relatively known. Bork et al. showed that $SO_3^-$ can form $SO_3$, the precursor for $H_2SO_4$, through electron transfer to ozone ($O_3$) (Bork et al., 2012). $SO_3^-$ can equally react with $O_2$ to form $SO_5^-$ whose atmospheric outcome by reaction with $O_3$ is $H_2SO_4$ formation (Bork et al., 2013). It was also speculated from chamber studies that $SO_5^-$ could form and stabilize clusters with sulfuric acid in the gas-phase (Kirkby et al., 2011). Reliable predictions of the outcomes of these ions require an exact knowledge of their chemical structures since interactions between molecules or ions depend both on their physical and chemical properties. A previous study demonstrated that two forms of $SO_4^-$ separated by a high energy barrier may exist in the atmosphere (Tsona et al., 2014): the sulfate radical ion henceforth indicated as $SO_4^-$, and the peroxy form, $O_2SOO^-$, in which the $O_2S–OO^-$ bond nature is more dative than covalent. Formerly, the two isomers were often misleadingly attributed exclusively to the sulfate radical ion, the most stable form of $SO_4^-$. However, their reactive properties greatly differ (Fehsenfeld and Ferguson, 1974).

The formation mechanisms of $SO_4^-$ in the gas-phase have been largely unknown but, recent studies showed that this ion can be formed by $SO_5^-$ reaction with $O_3$ (Bork et al., 2013) and in a $O_2SOO^-$ isomerization process catalyzed by NO (Tsona et al., 2018). $SO_4^-$ can also be produced during the chemical transformation of organic compounds, triggered by sulfate salts (Noziere et al., 2010), whereas $O_2SOO^-$ is formed at collision rates upon $SO_2$ reaction with $O_2^-$ (Fahey et al., 1982; Tsona et al., 2014). The sulfate radical ion is believed to react with unsaturated compounds to form organosulfates, a major component of secondary organic aerosol (Surratt et al., 2007; Surratt et al., 2008; Schindelka et al., 2013). Using first-principles calculations, $SO_4^-$ was shown to act as a catalyst in $SO_2$ oxidation to $SO_3$ by $O_3$ in the gas-phase and hence, plays a role in atmospheric aerosol formation (Tsona et al., 2015; Tsona et al., 2016). The chemistry of $O_2SOO^-$ is largely unknown, although potentially important for sulfur chemistry and atmospheric aerosol formation. Fehsenfeld and Ferguson found that $O_2SOO^-$ can be decomposed by $NO_2$ into $NO_3^-$ and $SO_3$ (Fehsenfeld and Ferguson, 1974) and a recent study showed that in the presence of nitrogen oxides ($NOx = NO_2 + NO$), $O_2SOO^-$ can be converted into sulfate (Tsona et al., 2018). In mildly polluted environments, the concentration of $O_3$ can be few orders of magnitude higher than that of $NOx$ and the chemical fate of $O_2SOO^-$ would then also greatly depend on collisions with $O_3$. In such environments, $O_2SOO^-$ could experience much more collisions with $O_3$ than with $NOx$.

Hereby, we investigate the reaction between $O_2SOO^-\cdots(H_2O)_{0-1}$ and $O_3$ using *ab initio* calculations. By determining the reactions thermodynamics and kinetics, we examine the possible pathways for the reaction and propose the most probable outcome of $O_2SOO^-\cdots(H_2O)_{0-1}$ based on $O_3$ reaction. Implications of the most relevant pathways in aerosol formation are discussed.

## 2 Methods

### 2.1 Geometry optimizations, thermochemical and charge analysis

As the substrate in this study is a radical anion, all stationary points in the energy surface were optimized using density functional theory (DFT) based on the UM06-2X density functional (Zhao and Truhlar, 2008) and the aug-cc-pVTZ basis set (Dunning Jr et al., 2001), setting the charge to -1 and the spin multiplicity to 2. The use of UM06-2X implies using unrestricted wavefunctions to describe the quantum state of the system. Spin contamination often arises from unrestricted density functional theory (DFT) calculations and it is not guaranteed that the electronic states from these calculations are eigenstates of the $\hat{S}^2$ operator. The spin contamination was then evaluated for all electronic states as $\Delta S = \langle \hat{S}^2 \rangle - \langle \hat{S}^2 \rangle_{ideal}$, where $\langle \hat{S}^2 \rangle$ is actual value of the expectation value of the $\hat{S}^2$ operator from DFT calculations and $\langle \hat{S}^2 \rangle_{ideal}$ is the ideal expectation value. For systems explored in this study, $\langle \hat{S}^2 \rangle_{ideal} = 0.75$.

The UM06-2X functional has successfully proven to be adequate for reactions involving transition state (TS) configurations (Elm et al., 2012, 2013a, b). Harmonic vibrational frequencies analysis on the optimized structures were performed (at 298 K and 1 atm) using the UM06-2X/aug-cc-pVTZ method under the harmonic oscillator-rigid rotor approximation. These calculations ensured that the obtained stationary points were minima or TS and, also, provided the thermal contributions to the Gibbs free energy and the enthalpy.

Transition states structures were initially located by scanning the reactants configurations. The best TS guesses out of the scans were then refined using the synchronous transit quasi Newton method (Peng et al., 1996), and the final TS structures underwent internal reaction coordinate calculations (Fukui, 1981) to ensure they connected the reactants to desired products.

The electronic energies of the UM06-2X/aug-cc-pVTZ optimized geometries were corrected with the UCCSD(T) method (Purvis and Bartlett, 1982) in conjunction with the aug-cc-pVTZ basis set. The Gibbs free energies, $G$, of all relevant species were then calculated as

$$G = E_{\text{UCCSD(T)}} + G_{\text{therm}} \tag{1}$$

where $E_{\text{UCCSD(T)}}$ is the electronic energy calculated with the UCCSD(T)/aug-cc-pVTZ method and $G_{\text{therm}}$ is the thermal contribution to the Gibbs free energy, calculated at the UM06-2X/aug-cc-pVTZ level of theory. All geometry optimizations, harmonic vibrational frequencies analysis and electronic energies correction calculations were carried out in the Gaussian 09 package (Frisch et al., 2016).

To analyse the distribution of the excess electronic charge over different species and fragments in the optimized systems, we used the Atoms-in-Molecules charge partitioning method presented by Bader (Bader, 1998). This is an intuitive way of dividing the molecules of a system into atoms, which are purely defined in terms of electronic charge density. The Bader charge partitioning assumes that the charge density between atoms of a molecular system reaches a minimum, which is an ideal place to separate atoms from each other. As input, this method requires electronic density and nuclear coordinates from electronic structure calculations. We used the approach implemented in the algorithm developed by Henkelman and co-workers, which has been shown to work well both for charged and water-containing systems (Tang et al., 2009; Bork et al., 2011; Henkelman et al., 2006).

## 2.2 Reactions kinetics

Regardless of the presence of water, the reaction between $O_2SOO^-$ and $O_3$ begins by forming different van der Waals pre-reactive intermediates, depending on the orientation of the reactants at impact. The pre-reactive intermediate could either decompose to different species or react further through a transition state configuration to form new products:

$$O_2SOO^- + O_3 \leftrightarrow \text{Pre-reactive intermediate} \rightarrow \text{Products} \tag{R1}$$

The traditional approach to determine the rate constant of reaction (R1) relies on the steady-state approximation and leads to the following equation:

$$k = k_{\text{coll}} \frac{k_{\text{reac}}}{k_{\text{reac}} + k_{\text{evap}}} \tag{2}$$

where $k_{\text{coll}}$ is the collision frequency for $O_2SOO^-$-$O_3$ collisions, $k_{\text{evap}}$ is the rate constant for the evaporation of the pre-reactive intermediate back to initial reactants, and $k_{\text{reac}}$ is the unimolecular rate constant for the reaction of the pre-reactive intermediate to the products. Moreover, assuming that $k_{\text{evap}} \gg k_{\text{reac}}$, the rate constant of reaction (R1) becomes $k = K_{\text{eq}}k_{\text{reac}}$ over a range of temperatures, with $K_{\text{eq}}$ being the equilibrium constant of formation of the pre-reactive intermediate from the reactants. This consideration is, however, not valid for reactions with submerged barrier, since the pre-reactive intermediate seldom thermally equilibrates. For such reactions, a two-transition state theory has been introduced, treating two distinct transition state bottlenecks that define the net reactive flux (Klippenstein et al., 1988; Georgievskii and Klippenstein, 2005; Greenwald et al., 2005). The first bottleneck, the "outer" transition state, occurs in the association of the initial reactants to form the pre-reactive intermediate, whereas the second bottleneck, the "inner" transition state, occurs in the transformation of the pre-reactive intermediate to the products. Based on this theory, the overall rate constant ($k$) for a reaction channel is expressed in terms of the outer ($k_{\text{out}}$) and inner ($k_{\text{in}}$) rate constants as:

$$\frac{1}{k} = \frac{1}{k_{\text{out}}} + \frac{1}{k_{\text{in}}} \tag{3}$$

The outer transition state is treated by the long-range transition state theory approach (Georgievskii and Klippenstein, 2005), while the inner transition state is resolved by the transition state theory. It was shown that for interactions between ions and neutral molecules, due to their long-range attraction, the collision cross section is larger than would be measured from the physical dimensions of the colliding species (Kupiainen-Määttä et al., 2013). To account for this phenomenon, the outer rate constant was determined from the ion-dipole parametrization of Su and Chesnavich who performed trajectory simulations of collisions between a point charge and a rigidly rotating molecule (Su and Chesnavich, 1982). This parametrization is equivalent to a Langevin capture rate constant ($k_L$) scaled by a temperature-dependent term and was found to provide good agreement with experiments (Kupiainen-Määttä et al., 2013). It is given as

$$k_{\text{out}} = k_L \times \left( \frac{(x + 0.5090)^2}{10.526} + 0.9754 \right) \tag{4}$$

where $k_L = q\mu^{-1/2}(\pi\alpha/\varepsilon_0)^{1/2}$, $x = \mu_D/(8\pi\varepsilon_0\alpha k_B T)^{1/2}$, $q$ is the charge of the ion, $\mu$ is the reduced mass of the colliding species, $\varepsilon_0$ is the vacuum permittivity, $\alpha$ and $\mu_D$ are the polarizability and dipole moment of the neutral molecule (ozone), $k_B$ is Boltzmann's constant, and $T$ is the absolute temperature. The inner rate constant can be written as:

$$k_{in} = \frac{k_B T}{hc^0} \times \exp\left(-\frac{\Delta G^{\#}}{RT}\right) \tag{5}$$

where $\Delta G^{\#}$ is the Gibbs free energy barrier separating the pre-reactive intermediate and the products, $h$ is the Planck's constant, $R$ is the molar gas constant, and $c^0$ is the standard gas-phase concentration. The constants and variables in Eq. (4) and Eq. (5) are given in centimetre-gram-second (CGS) system of units and International System (SI) units, respectively. Details on these units are given in the Supplement.

## 3 Results and discussion

Starting with optimized structures of $O_2SOO^-\cdots(H_2O)_{0-1}$ and $O_3$ shown in Fig. S1, a series of geometry optimizations were performed on the $O_2SOO^-\cdots(H_2O)_{0-1} + O_3$ system, taking into account different spatial orientations of the reactants at impact. These optimizations led to two main chemical processes, depending on the initial orientation of the reactants, with potentially different outcomes. The first process is the formation of a van der Waals complex followed by its direct decomposition to other species. The second process is the low-lying formation of a molecular complex in which the $SO_2$ entity of $O_2SOO^-\cdots(H_2O)_{0-1}$ is oxidized to $SO_3^-$.

### 3.1 Cluster formation and decomposition of $O_2SOO^-\cdots(H_2O)_{0-1}$

As $O_3$ approaches $O_2SOO^-\cdots(H_2O)_{0-1}$ towards the oxygen atoms of the peroxy fragment or the oxygen atom of the $SO_2$ moiety, the immediate outcome of $O_2SOO^-\cdots(H_2O)_{0-1}$ and $O_3$ collisions is the formation of the van der Waals $O_3\cdots O_2SOO^-\cdots(H_2O)_{0-1}$ complex in which $O_3$ interacts with $O_2SOO^-$. Among the different stable configurations found upon optimizations, we solely report the most stable one with respect to the Gibbs free energy, which is henceforth denoted RC1 and RCW1 for the unhydrated and monohydrated, respectively, shown in Fig. 1. Exploration of the RC1 and RCW1 structures reveals that $O_2SOO^-\cdots(H_2O)_{0-1}$ basically keeps its configuration upon clustering with $O_3$. The spin contaminations for RC1 and RCW1 are negligible, being 0.0086 and 0.0081, respectively. The electronic energies of formation of RC1 and RCW1 are -5.1 and -4.6 kcal mol$^{-1}$, respectively. Despite these complexes may form at 0 K, the Gibbs free energies of their formation under atmospheric pressure and 298 K (4.5 and 4.7 kcal mol$^{-1}$, respectively) indicate that their formation is endergonic under atmospherically relevant conditions. These Gibbs free energy values indicate that, if formed, these complexes would not live long and will rather decompose either to initial reactants or to different species. Hence, the Gibbs free energies of formation of RC1 and

RCW1 define the lowest states at which $O_2SOO^-$ can interact with $O_3$ to allow electron transfer and $O_2S–OO$ decomposition, and thus represent the energy barrier towards $O_2 + SO_2 + O_3^-$ formation. Inspecting the vibrational modes of RC1 and RCW1, two vibrations are found that would clearly lead to the dissociation of $O_2SOO^-$ within the complex. The analysis of the charge distribution over $O_3\cdots O_2SOO^-\cdots(H_2O)_{0-1}$ shows that the extra electron initially located on $O_2SOO$ in the reactants has migrated to the $O_3$ molecule in the van der Waals product complex, as can be observed in Fig. S2. This is as expected, given the high electronegativity of $O_3$ relative to those of $O_2$ and $SO_2$ (Rothe et al., 1975). The charge distribution over the different atoms of the optimized complex is weakly affected by the presence of water, as previously demonstrated by Bork and co-workers (Bork et al., 2011).

The most likely fates of RC1 and RCW1 are, therefore, decomposition into $O_2$, $SO_2$ and $O_3^-$ as follows:

$$O_2SOO^-\cdots(H_2O)_{0-1} + O_3 \leftrightarrow O_3\cdots O_2SOO^-\cdots(H_2O)_{0-1} \rightarrow O_2 + SO_2 + O_3^- + (H_2O)_{0-1} \qquad (R2)$$

The numerical values of the formation energies of all intermediate species in reaction (R2) are given in Table 1 and the energy surfaces are plotted in Fig. 2. RC1 and RCW1 decompositions are highly exergonic at 298 K, occurring with -18.1 and -16.7 kcal mol$^{-1}$ Gibbs free energy changes, respectively. These processes are, therefore, likely to occur in the atmosphere upon formation of $O_3\cdots O_2SOO^-\cdots(H_2O)_{0-1}$.

The limiting step in reaction (R2) is the formation of RC1 and RCW1, whose formation energies indicated above can then be considered as the only barrier to the formation of $O_2 + SO_2 + O_3^-$. This leads to overall rate constants (according to Eq. 5) of $1.4\times10^{-10}$ and $9.9\times10^{-11}$ cm$^3$ molecule$^{-1}$ s$^{-1}$ at 298 K for the unhydrated and monohydrated reaction, respectively. Both reactions are, in principle, collision-limited and the effect of hydration on the kinetics is found to be negligible. The atmospheric relevance of reaction (R2) has been determined earlier (Bork et al., 2012; Bork et al., 2013; Enghoff et al., 2012).

### 3.2 $O_2SOO^-\cdots(H_2O)_{0-1}$ reaction with $O_3$

When $O_3$ approaches $O_2SOO^-\cdots(H_2O)_{0-1}$ from the sulfur atom side, the formation of a more stable cluster than found above prevails. The incoming $O_3$ molecule strongly interacts with $O_2SOO^-\cdots(H_2O)_{0-1}$ by forming a coordination bond with the sulfur atom and hereby, inducing the ejection of the $O_2$ molecule that remains in interaction with the remainder of the system. This process leads to the formation of the $O_2\cdots O_2S–O_3^-\cdots(H_2O)_{0-1}$ complex which further transforms, through an intramolecular $SO_2$ oxidation, into $SO_3^-\cdots(H_2O)_{0-1} + 2O_2$ according to the following equation:

$$O_2SOO^-\cdots(H_2O)_{0-1} + O_3 \leftrightarrow O_2\cdots O_2S–O_3^-\cdots(H_2O)_{0-1} \rightarrow SO_3^-\cdots(H_2O)_{0-1} + 2O_2 \qquad (R3)$$

The configurations of the most stable intermediate structures in reaction (R3) are given in Fig. 1. The charge analysis on this system indicates that the electronic charge on the pre-reactive complex is essentially on two oxygen atoms of the $O_3$ moiety that is coordinated to $SO_2$ as can be seen in Fig. 3(a). The net charge on these two oxygen atoms is 1.04e, whereas the net charge on the free $O_2$ molecule is 0.01e. The latter value shows that the $O_2$ molecule formed in the pre-reactive complex has no unpaired electrons, and hence is a singlet. Although the charge is still on the $O_3$ moiety in the transition state configuration, it is mostly located on the oxygen atom bound to sulfur (**Fig. 3(b)**). The net charge on the two outer oxygen atoms of the $O_3$ moiety in the transition state has substantially decreased to 0.30e while the charge on the free $O_2$ molecule has slightly increased to 0.04e. The strong decrease in the charge of the two outer oxygen atoms of $O_3$ from the pre-reactive complex to the transition state suggests that the $O_2$ molecule to form in the product will likely be a singlet. In the products (**Fig. 3(c)**), the old free $O_2$ molecule has a net charge of 1.99e, whereas the charge on the newly formed $O_2$ is 0.06e. The 1.99e charge on the old free $O_2$ molecule indicates the presence of unpaired electrons in its configuration, meaning that the singlet $O_2$ has been transformed into triplet. This clearly shows that a spin flip has occurred in $O_2$ during further optimization of the products. The newly formed $O_2$ with 0.06e charge is obviously a singlet. This analysis shows for the unhydrated system that the singlet $O_2$ initially formed in the pre-reactive complex transforms into triplet in the products state, while a new singlet $O_2$ is also formed. In the monohydrated system, the singlet $O_2$ initially formed in the pre-reactive complex remains as singlet in the products state and a triplet $O_2$ is also released. Overall, the two forms of $O_2$ (singlet and triplet) are formed in the studied reaction, despite following different mechanisms. Although the water molecule in the monohydrated system does not retain any electric charge, it most likely stabilizes the initially formed singlet $O_2$ and prevents the spin flip.

Though the necessity to determine the electronic structure of the $O_2$ molecule in the singlet sate ($^1\Delta_g$) has been demonstrated to be useful (Buttar and Hirst, 1994), obtaining a reliable electronic energy for $O_2(^1\Delta_g)$ is difficult (Drougas and Kosmas, 2005). An alternative approach to determine this energy is to add the experimental energy spacing (22.5 kcal mol$^{-1}$) between triplet and singlet states of $O_2$ to the computed electronic energy of the triplet $O_2$ (Schweitzer and Schmidt, 2003; Drougas and Kosmas, 2005). We used this approach to determine the energies of the products of reaction (R3). The numerical values of the formation energies of all intermediate species in reaction (R3) are thus given in Table 1 and the energy surfaces are plotted in Fig. 2. The most stable optimized structures of $O_2\cdots O_2S-O_3^-\cdots(H_2O)_{0-1}$ according to our calculations are denoted as RC2 and RCW2 for the unhydrated and monohydrated systems, respectively and are shown in Fig. 1. Regardless of the presence of water, the $O_2\cdots O_2SO_3^-$ configuration results from $O_2$ being switched by $O_3$ in the $O_2SOO^-$ molecular ion. In the optimized $O_2\cdots O_2S-O_3^-$ structure, the O3 atom of $O_3$ points towards the S atom of $O_2SOO^-$, forming S–O3 bonds at distances of 1.90 and 1.87 Å in the absence and presence of water, respectively. These bonds are coordination bonds in nature since the S-O covalent bond in e.g., 1.43 Å in $SO_3$ and 1.46 Å in $H_2SO_4$. The S-O3 coordination bond distances in RC2 and RCW2 are shorter by 0.04 and 0.03 Å than $O_2S-OO^-$ bond distances in unhydrated and monohydrated $O_2SOO^-$ forms. This indicates stronger interaction between $O_3$ and $SO_2$, and hence higher stability of $O_2\cdots O_2S-O_3^-$ relative to $O_2SOO^-$.

The formations of RC2 and RCW2 are highly exergonic, with Gibbs free energy changes at 298 K of -14.7 and -12.4 kcal mol$^{-1}$, respectively. These values, with corresponding electronic energies and enthalpies are shown in Table 1. These Gibbs free energy changes for the formation of RC2 and RCW2 are about 19 kcal mol$^{-1}$ lower than those of RC1 and RCW1 at similar conditions, indicating the higher stability of RC2 and RCW2, and the highly favourable switching reaction at ambient conditions. $SO_2$ oxidation can readily occur within the $O_2 \cdots O_2S-O_3^- \cdots (H_2O)_{0-1}$ cluster and lead to $SO_3^-$ formation. In principle, to form the products of reaction (R3), the O3 atom of the $O_3$ moiety in $O_2 \cdots O_2S-O_3^- \cdots (H_2O)_{0-1}$ transfers, through transition state configurations, to $SO_2$ and form $SO_3^-$ followed by the ejection of the $O_2$ molecule. The transition states are denoted TS2 and TSW2 for the unhydrated and monohydrated systems, respectively, and their structures are presented in Fig. 1. While RC2 and RCW2 are formed with similar Gibbs free energies within 2 kcal mol$^{-1}$ difference, the formation Gibbs free energies of their transition states at similar conditions greatly differ. TS2 is located at 10 kcal mol$^{-1}$ Gibbs free energy above RC2, while TSW2 is located at -4 kcal mol$^{-1}$ below RCW2. It is speculated that the low energy barrier in the monohydrated reaction is the result of a strong stabilisation of the transition state due to hydration, with the S-O3 bonds in RCW2 and TSW2 shorter by ~0.03 Å than in RC2 and TS2. Another reason for this substantial drop in energy barrier is the difference in the electronic configurations of the two outer oxygen atoms of the $O_3$ moiety in the two transition states that form $O_2$ with different multiplicities in the products.

Based on the TS2 energy, the unimolecular decomposition of $O_2 \cdots O_2S-O_3^-$ at 298 K in the absence of water was found to occur at a rate constant of $3.1 \times 10^5$ s$^{-1}$, corresponding to an atmospheric lifetime of 3.3 µs. Both this short lifetime and the negative energy barrier of the monohydrated reaction indicate that $O_2 \cdots O_2S-O_3^-$ would not live long enough to experience collisions with other atmospheric oxidants. It should be noted that few to no collisions with nitrogen can, however, be achieved. It follows that the most likely outcome of $O_2 \cdots O_2S-O_3^-$ is decomposition to the products of reaction (R3), which are formed with about -46 kcal mol$^{-1}$ overall Gibbs free energy at 298 K. The net reaction is an $O_2^-$-initiated $SO_2$ oxidation to $SO_3^-$ by $O_3$.

The spin contamination for electronic states in reaction (R3) is quite significant, being 1.0122, 1.4666, and 2.0374 for the pre-reactive complex, transition state and product, respectively, and is almost insensitive to the presence of water. The actual values of the expectation values of the $\hat{S}^2$ operator for all electronic states obtained from our calculations are given in the Supplement, along with their cartesian coordinates. The high values of spin contamination likely reflect the formation of $O_2$ with different multiplicities within the system. As the charge analysis indicates, starting with singlet $O_2$ in the pre-reactive complexes of reaction (R3), both singlet and triplet $O_2$ are formed in the final products.

The overall rate constants of reaction (R3), determined at 298 K using Eq. (3), are $1.3 \times 10^{-14}$ and $8.0 \times 10^{-10}$ cm$^3$ molecule$^{-1}$ s$^{-1}$ for the unhydrated and monohydrated reactions, respectively. The values of the different components ($k_{out}$ and $k_{in}$) are listed in Table S1 of the Supplement. It is observed from Table S1 that the inner transition sate provides the dominant bottleneck to the

rate constant of the unhydrated reaction, whereas the outer transition state provides the dominant bottleneck to the rate constant of the monohydrated reaction.

The effective effect of water on the rate constant can be evaluated by taking into account the stability of $O_2SOO^{-}\cdots H_2O$ (which is formed at the entrance channel of the reaction in the presence of water before colliding with $O_3$) and the equilibrium vapor pressure of water. Starting from the definition of the reaction rate for the hydrated reaction,

$$J_{(R3w)} = k_{(R3w)} \times [O_2SOO^{-}\cdots H_2O] \times [O_3] \tag{6}$$

$$= k_{(R3w)}^{eff} \times [O_2SOO^{-}] \times [O_3] \tag{7}$$

where $k_{(R3w)}$ is the overall rate constant for the hydrated reaction, determined using Eq. (3), $k_{(R3w)}^{eff}$ is the effective reaction rate constant calculated as $k_{(R3w)}^{eff} = k_{(R3w)} \times K_{eq} \times p_{H2O}$. $K$eq is the equilibrium constant for the $O_2SOO^{-} + H_2O \leftrightarrow O_2SOO^{-}\cdots H_2O$ reaction and $p_{H2O}$ the actual water vapor pressure. Details on the determination of $K_{eq}$ and $p_{H2O}$ are given in the Supplement.

At 298 K and 50 % relative humidity, the effective rate constant of the monohydrated reaction is $1.7 \times 10^{-10}$ cm$^3$ molecule$^{-1}$ s$^{-1}$, four orders of magnitude higher than the rate constant of the unhydrated reaction. Therefore, water plays a catalytic role on the kinetics of reaction (R3). The net rate constant of reaction (R3) can be obtained by weighing the rate constants of the unhydrated and monohydrated reactions to corresponding equilibrium concentrations of $O_2\cdots O_2S–O_3^{-}$ hydrates. Using the law of mass action, we find that $O_2\cdots O_2S–O_3^{-}$ mostly exists as a dry species, constituting 77% of the total population, whereas the monohydrated species forms 23% of the total population. The net rate constant of reaction (R3) is then determined to be $4.0 \times 10^{-11}$ cm$^3$ molecule$^{-1}$ s$^{-1}$ at 298 K.

Considering only the unhydrated process of reaction (R3), the rate constant is 4-5 orders of magnitude lower than the rate constant obtained for the $SO_2 + O_3^{-} \rightarrow SO_3^{-} + O_2$ reaction (Fehsenfeld and Ferguson, 1974; Bork et al., 2012). Despite this difference, the oxidation process follows a similar mechanism to the one presented by Bork et al. for the $SO_2 + O_3^{-} \rightarrow SO_3^{-} + O_2$ reaction, consisting of the oxygen transfer from $O_3$ to $SO_2$ (Bork et al., 2012). The discrepancy between the two results is associated with the effect of the presence of the O–O fragment initially coordinated to $SO_2$ in the current study, which tends to stabilize the $O_2\cdots O_2S–O_3^{-}$ pre-reactive complex. The presence of the O-O fragment seemingly deactivates $SO_2$ for the upcoming O transfer from $O_3$ to form $SO_3^{-}$. However, this situation is rapidly reversed with the presence of water as the reaction becomes much faster, proceeding nearly at collision rate.

## 3.3 Further chemistry

In real atmospheric and ionized conditions, despite $O_2$ has lower electron affinity than $O_3$, it would likely ionize faster than $O_3$ owing to its much higher concentration. Considering for example chamber experiments, upon interaction of ionizing particles with the gas, electrons can transfer from one species to another and, e.g., $O_2^-$ can form and rapidly hydrate within one nanosecond (Svensmark et al., 2007; Fahey et al., 1982). Furthermore, Fahey et al. showed that $O_2^- \cdots (H_2O)_{0-1}$ association reaction with $SO_2$ is faster than the electron transfer from $O_2^- \cdots (H_2O)_{0-1}$ to $O_3$ (Fahey et al., 1982). This means that in an ionized environment containing $O_2$, $O_3$, and $SO_2$, the formation of $O_2S-OO^-$ resulting from $SO_2$ and $O_2^-$ association will happen faster than $O_3^-$ formation. $O_2S-OO^-$ would react thereafter with $O_3$ and the following stepwise process could take place

$$O_2^- \cdots (H_2O)_{0-1} + SO_2 \rightarrow O_2S-OO^- \cdots (H_2O)_{0-1} \tag{R4a}$$

$$O_2S-OO^- \cdots (H_2O)_{0-1} + O_3 \rightarrow O_2 \cdots O_2S-O_3^- \cdots (H_2O)_{0-1} \tag{R4b}$$

$$O_2 \cdots O_2S-O_3^- \cdots (H_2O)_{0-1} \rightarrow SO_3^- \cdots (H_2O)_{0-1} + 2O_2 \tag{R4c}$$

$$\text{Net: } O_2^- \cdots (H_2O)_{0-1} + SO_2 + O_3 \rightarrow SO_3^- \cdots (H_2O)_{0-1} + 2O_2 \tag{R4}$$

The Gibbs free energy change of this net reaction at 298 K is about -40 kcal $mol^{-1}$ more negative than that of the $SO_2 + O_3^- \rightarrow SO_3^- + O_2$ reaction at similar conditions. Given that the intermediate steps of reaction (R4) are significantly fast, this reaction is believed to be an important process in most environments of $SO_2$ ion-induced oxidation to $SO_3^-$ or more oxidized species. The limiting step in the process of reaction (R4) is reaction (R4c) for which an energy barrier has to be overcome before the products are released.

$SO_3^-$ is an identified stable ion detected in the atmosphere and in experiments (Ehn et al., 2010; Kirkby et al., 2011; Kirkby et al., 2016). The chemical fate of $SO_3^-$ is fundamentally different from that of $SO_3$ that forms $H_2SO_4$ by hydration. Likely outcomes of $SO_3^-$ are hydrolysis, electron transfer by collision with $O_3$, reaction with $O_2$ and $H_2O$ and, possibly, radicals, according to the following equations

$$SO_3^- + H_2O \rightarrow SO_3^- \cdots H_2O \rightarrow H_2SO_4 + e^- \tag{R5}$$

$$SO_3^- + O_3 \rightarrow SO_3 + O_3^- \tag{R6}$$

$$SO_3^- + O_2 + H_2O \rightarrow HSO_4^- + HO_2 \tag{R7}$$

$$SO_3^- + OH \rightarrow HSO_4^- \tag{R8}$$

Fehsenfeld and Ferguson showed that $H_2SO_4$ formation could occur in the $SO_3^- \cdots H_2O$ cluster, releasing a free electron (Reaction (R5)) (Fehsenfeld and Ferguson, 1974). Owing to the high electron affinity of $O_3$ relative to $SO_3$ (Rothe et al., 1975), the electron can transfer from $SO_3$ to $O_3$ and lead to the formation of $SO_3$, the precursor for sulfuric acid in the atmosphere.

Moreover, the free electron released and the $O_3^-$ formed in reactions (R5) and (R6), respectively, are potential triggers of new $SO_2$ oxidations with implication in aerosol formation (Svensmark et al., 2007; Enghoff and Svensmark, 2008; Bork et al., 2013). Reactions (R7) and (R8) are potential outcomes for $SO_3^-$ as well, forming the highly stable $HSO_4^-$ species that would terminate the oxidation process of $SO_2$ initiated by a free electron. Reactions (R5)–(R8) are likely competitive processes upon

$SO_3^-$ formation in the gas-phase, and their different rates would determine the number of $SO_2$ oxidations induced by a free electron. However, they have no other fate than $HSO_4^-$ or $H_2SO_4$, the most oxidized sulfur species in the atmosphere, which both share many properties and play a central role in atmospheric particle formation.

Experimental studies have shown that in atmospheres heavily enriched in $SO_2$ and $O_3$, a free electron could initiate $SO_2$ oxidation and induce the formation of $\sim 10^7$ cm$^{-3}$ sulfates in the absence of UV light, clearly indicating the importance of other

ionic $SO_2$ oxidation mechanisms than UV-induced (Enghoff and Svensmark, 2008). To evaluate the importance of the mechanism presented in this study in the formation of sulfate, it is necessary to identify the scavengers that terminate the $SO_2$ oxidation initiated by $O_2^-$. Possible scavengers include radicals, NOx, acids, cations and other particles. The main ones are likely NOx, OH, $HO_2$ and organic acids, which lead to the formation of the stable $NO_3^-$, $HSO_4^-$, and $CO_3^-$ species. If the ion concentration was known, the contribution of reaction (R4) to $H_2SO_4$ formation could be determined by comparing its

formation rates from ionic and electrically neutral mechanisms. Alternatively, it can be assumed that reaction (R4) is terminated when the ion cluster hits a scavenger. The free electron which acts as catalyst is then scavenged. The average catalytic turnover number (TON) is defined as (Kozuch and Martin, 2012):

$$TON = \frac{\text{concentration of limiting reacted molecules}}{\text{concentration of the catalyst}} \qquad (8)$$

The concentration of the catalyst can be approximated to the concentration of the scavengers and, considering that at most atmospheric conditions $[O_3] > [SO_2]$, $SO_2$ is the limiting species in reaction (R4). Equation (8) can then be re-written as

$$TON \approx \frac{[SO_2]}{[OH] + [HO_2] + [NO_x] + [\text{organic acids}]} \qquad (9)$$

The catalytic efficiency of $SO_2$ ion-induced oxidation is then given as

$$J_{ion} = k_{ion} \times TON \qquad (10)$$

Where $k_{ion}$ is the ion production rate. Depending on the tropospheric temperature and altitude, measurements at the Cosmics Leaving Outdoor Droplets chamber at CERN found $k_{ion} = 2$–$100$ cm$^{-3}$ s$^{-1}$, covering the typical ionization range in the troposphere (Franchin et al., 2015). We assume nearly pristine conditions with $[SO_2] = 5$ ppb $= 1.2 \times 10^{11}$ molecule cm$^{-3}$, $[NOx] = 200$ ppt $= 4.9 \times 10^9$ molecule cm$^{-3}$, $[OH] = 5.0 \times 10^5$ molecule cm$^{-3}$ (day and night average), and $[HO_2] = 10^8$

molecule cm$^{-3}$ (Dusanter et al., 2009; Holland et al., 2003). Noting that formic acid and acetic acid are the most abundant organic acids in the atmosphere, their concentrations are considered in Eq. (9) as representative examples for organic acids, [organic acids] = 110 ppt = $2.7 \times 10^9$ molecule cm$^{-3}$ (Le Breton et al., 2012; Baasandorj et al., 2015). We then determine $J_{ion}$ in the range $3.2 \times 10^1$–$1.6 \times 10^3$ cm$^{-3}$ s$^{-1}$. The rate of the UV-induced SO$_2$ oxidation by OH is

$$J_{UV} = k_{UV} \times [SO_2] \times [OH] \tag{11}$$

With $k_{UV} = 1.3 \times 10^{-12}$ cm$^3$ molecule$^{-1}$ s$^{-1}$ (Atkinson et al., 2004), $J_{UV} = 7.9 \times 10^4$ molecule cm$^{-3}$ s$^{-1}$, and the proportion of H$_2$SO$_4$ formed from ion-induced oxidation can be estimated from the following equation

$$\frac{[H_2SO_4]_{ion}}{[H_2SO_4]_{total}} = \frac{J_{ion}}{J_{UV}+J_{ion}} \tag{12}$$

We find that the contribution of ion-induced SO$_2$ oxidation to H$_2$SO$_4$ formation can range from 0.1 to 2.0% of the total formation rate. This estimate could be improved by considering also the SO$_2$ oxidation by Criegee Intermediates, another important channel for H$_2$SO$_4$ formation.

## 3 Conclusions

This study highlights the role of the superoxide ions (O$_2^-$) in SO$_2$ oxidation. Our previous study demonstrated that SO$_2$ interacts with O$_2^-$ and forms O$_2$SOO$^-$ whose atmospheric fate remains unelucidated (Tsona et al., 2014). In this study, we used ab initio calculations to assess the chemical fate of O$_2$SOO$^-$ by collisions with O$_3$. Regardless of the presence of water, two main mechanisms are observed, leading to fundamentally different products. The first mechanism is characterized by electron transfer followed by O$_2$SOO$^-$ decomposition, leading to O$_3^-$ formation and releasing SO$_2$. The chemistry of SO$_2$ + O$_3^-$ has been explored elsewhere. The second mechanism is characterized by SO$_2$ oxidation and proceeds through formation of a pre-reactive complex that subsequently reacts to form the products by overcoming a relatively low energy barrier. The overall reaction, O$_2^-$ + SO$_2$ + O$_3$ → SO$_3^-$ + 2O$_2$, is faster and more energetically favourable than the SO$_2$ + O$_3^-$ → SO$_3^-$ + O$_2$ reaction, thereby highlighting the positive role of O$_2^-$ in SO$_2$ ionic oxidation. Hence, the two reactions may compete in chamber experiments and in the atmosphere.

While for the electron transfer and O$_2$SOO$^-$ decomposition process the reaction is hindered by the presence of water, the oxidation reaction is catalysed instead as the rate constant is increased by 6 orders of magnitude with the presence of water. Weighing the rate constants of unhydrated and monohydrated reactions to the equilibrium concentrations of hydrates of corresponding pre-reactive complexes leads to the net rate constant of $4.0 \times 10^{-11}$ cm$^3$ molecule$^{-1}$ s$^{-1}$ at 298 K for the oxidation reaction. Hence, this reaction proceeds nearly at collision rate. The main species (SO$_3^-$) in the end products of the studied

reaction has been proved to form both in the atmosphere and in experiments, where it definitely plays a role in atmospheric sulfur chemistry and particle formation. The contribution of this mechanism to the total atmospheric sulfuric acid formation is estimated. The studied reaction further deepens the understanding of ion-induced $SO_2$ oxidation, with implications in aerosol formation.

## 5 Author contributions

NTT and LD designed the work. NTT performed all calculations and analysed the data. NTT wrote the whole manuscript and LD edited it.

## Competing interests

The authors declare that they have no conflict of interest.

## 10 Acknowledgements

This work was supported by the National Natural Science Foundation of China (21707080, 91644214), the Postdoctoral Science Foundation of China (2017M612276), Shandong Natural Science Fund for Distinguished Young Scholars (JQ201705) and the International Postdoctoral Exchange Fellowship Program. We acknowledge the High-Performance Computing Center of Shandong University for providing the computational resources.

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

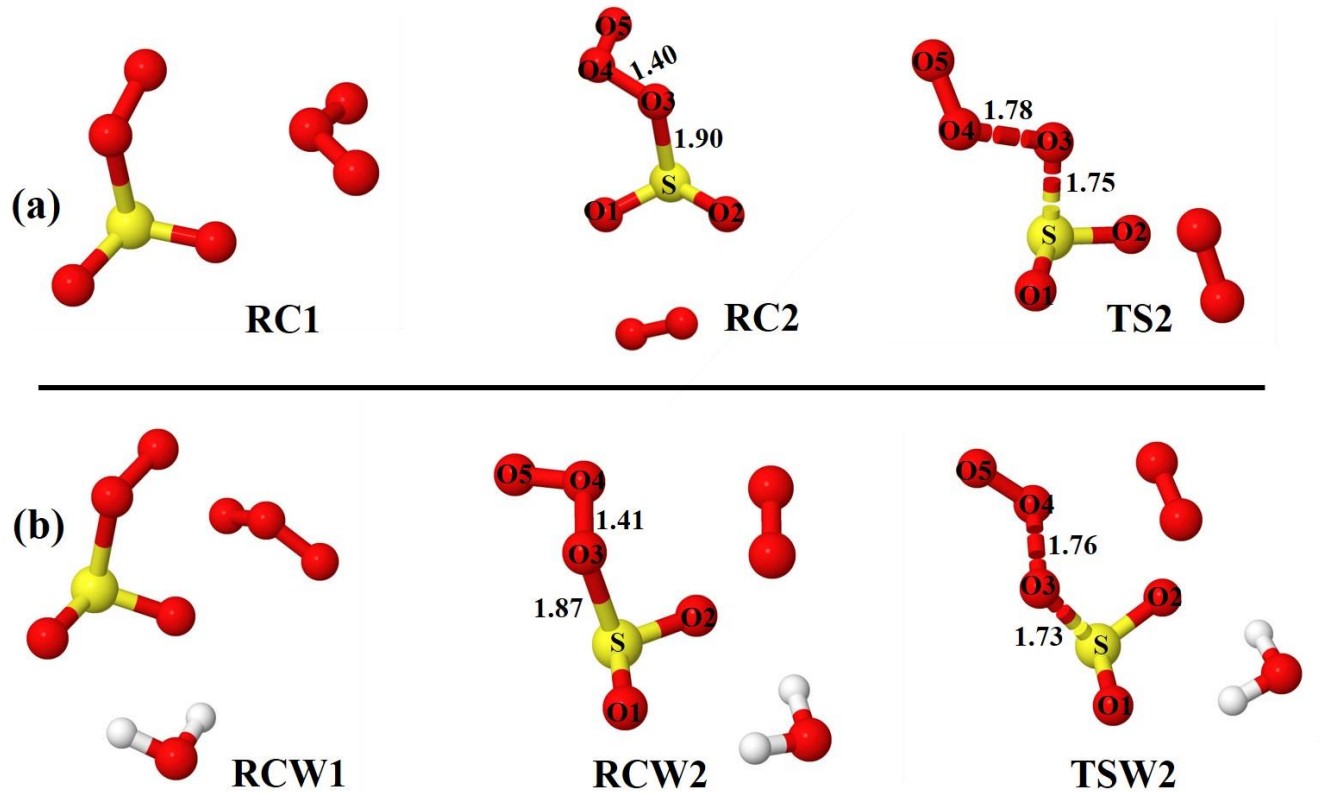

**Figure 1: Optimized structures of the most stable intermediates in the $O_2SOO^-$ + $O_3$ reaction (a) in the absence and (b) in the presence of a single water molecule. Optimizations were performed at the UM06-2X/aug-cc-pVTZ level of theory. Lengths (in Å) of some descriptive bonds are indicated. The color coding is yellow for sulfur, red for oxygen and white for hydrogen.**

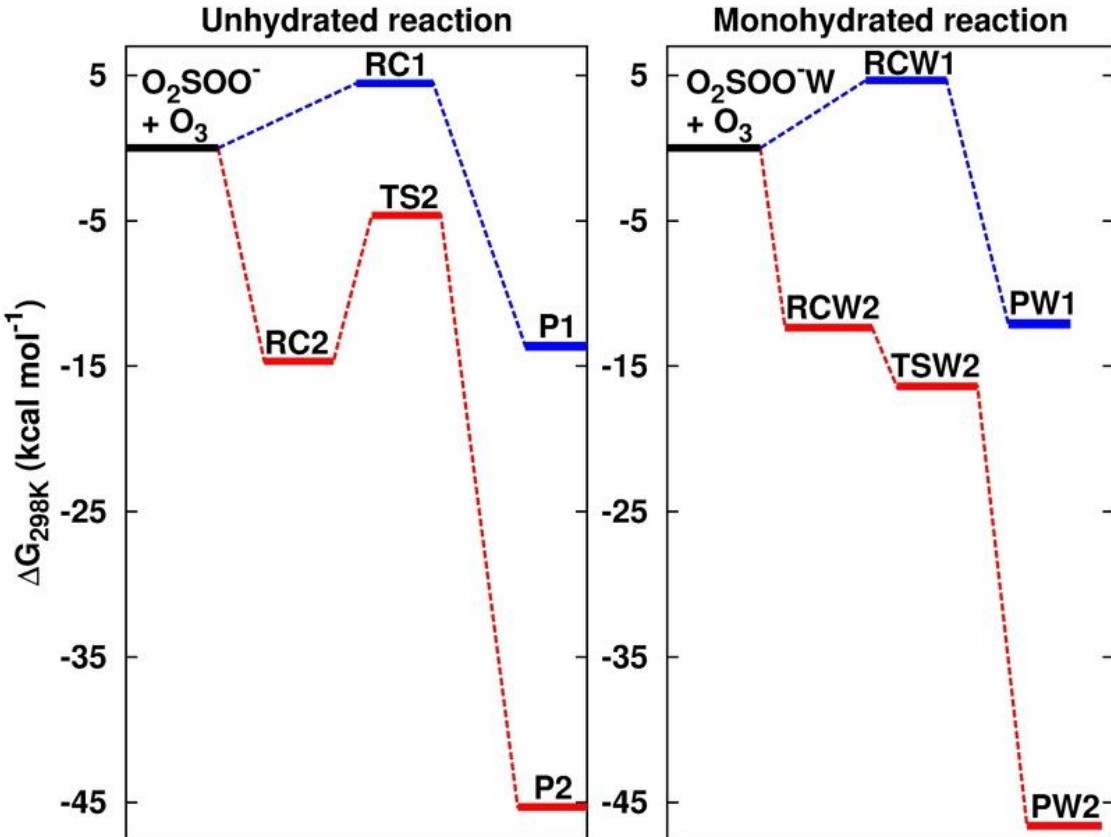

**Figure 2:** Formation Gibbs free energies of the most stable intermediate species in the $O_2SOO^- + O_3$ reaction in the absence and in the presence of water. "W" is the shorthand notation for water. RC1, RC2, TS2, RCW1, RCW2, and TSW2 structures are shown in Fig. 1. $P1 = O_2 + SO_2 + O_3^-$, $P2 = SO_3^- + 2O_2$, $PW1 = O_2 + SO_2 + O_3^- + H_2O$ and $PW2 = SO_3^- \cdots H_2O + 2O_2$. Calculations were performed at the UCCSD(T)/aug-cc-pVTZ//UM06-2X/aug-cc-pVTZ level of theory.

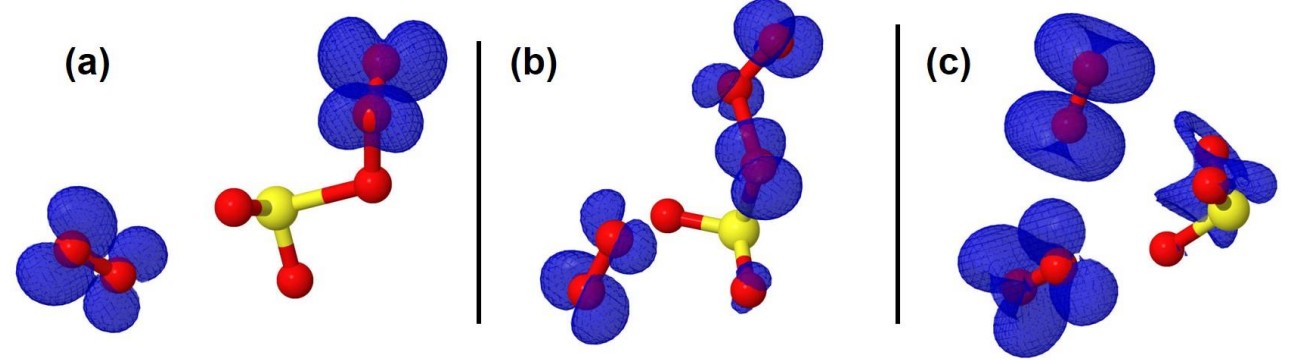

**Figure 3: Representation of the spin density (in blue color) on intermediate structures in the $O_2SOO^-$ + $O_3$ reaction. The spin density clearly indicates that the extra electron is progressively distributed over all the atoms from (a) the pre-reactive complex through (b) the transition state to (c) the product complex.**

**Table 1: Electronic energies ($\Delta E$), enthalpies ($\Delta H_{298K}$) and Gibbs free energies ($\Delta G_{298K}$) of the different states in the $O_2SOO^-$ + $O_3$ reaction both in the absence and in the presence of water, calculated relative to the energy of initial reactants at the UCCSD(T)/aug-cc-pVTZ//UM06-2X/aug-cc-pVTZ level of theory.**

| Species | $\Delta E$ | $\Delta H_{298K}$ | $\Delta G_{298K}$ |
|---|---|---|---|
| Unhydrated reaction | | | |
| $O_2SOO^-$ + $O_3$ | 0 | 0 | 0 |
| RC1 = $O_3\cdots O_2SOO^-$ | -5.1 | -3.9 | 4.5 |
| RC2 = $O_2\cdots O_2S-O_3^-$ | -21.9 | -21.0 | -14.7 |
| TS2 | -11.6 | -11.9 | -4.6 |
| $SO_3^-$ + $2O_2$ | -35.8 | -36.4 | -45.3 |
| $O_2$ + $SO_2$ + $O_3^-$ | -1.7 | -3.2 | -13.6 |
| Monohydrated reaction | | | |
| $O_2SOO^-\cdots H_2O$ + $O_3$ | 0 | 0 | 0 |
| RCW1 = $O_3\cdots O_2SOO^-\cdots H_2O$ | -4.6 | -3.4 | 4.7 |
| RCW2 = $O_2\cdots O_2S-O_3^-\cdots H_2O$ | -21.9 | -20.9 | -12.4 |
| TSW2 | -25.3 | -25.4 | -16.4 |
| $SO_3^-\cdots H_2O$ + $2O_2$ | -36.7 | -37.4 | -46.6 |
| $O_2$ + $SO_2$ + $O_3^-$ + $H_2O$ | 10.3 | 7.1 | -12.1 |