# Peer review of "A potential source of atmospheric sulfate from $O_2$ -induced $SO_2$ oxidation by ozone"

_Atmospheric Chemistry and Physics, 2018_

## Referee Comment (RC1) · Anonymous Referee #1 · 28 Nov 2018

Tsona and Du have studied the reaction of O3 with O2SO2- using computational methods. This extends and complements a series of similar studies performed by the authors and their co-workers, aimed toward understanding the ionic contribution to sulfur chemistry in the atmosphere. The computational methods are broadly appropriate to the task (see below for two minor caveats on this), and the results are interesting for atmospheric chemists despite the studied SO2 oxidation pathway ultimately being rather minor compared to neutral channels. Overall, the text is understandable, though there are a large number of odd word choices and formulations - some proofreading or copyediting would improve the manuscript. I recommend the manuscript be published in ACP subject so some minor revisions.

Comments:

-Page 2, Line 5-7: The authors list different oxidation routes for SO2 in the atmosphere: OH, sCIs, ions and mineral dust. The first three are gas-phase processes, the latter I assume corresponds to heterogeneous SO2 oxidation. However, there are many more heterogeneous pathways for SO2 oxidation, many of which are likely even more important than mineral dust, such as aqueous-phase oxidation inside cloud droplets. These could thus be mentioned.

-Page 2, Line 11: The role of ions in aerosol formation has indeed been well established, and that role is in most conditions essentially "real but small". I.e. ionic pathways make a non-negligible contribution to aerosols, but in most atmospheric conditions the neutral pathways still dominate. This could be mentioned.

-Page 3, Line 19: for sulfur-containing compounds, would it not be better/safer to use the aug-cc-pV(T+d)Z basis set instead of aug-cc-pVTZ? This should provide additional accuracy (especially for bond formation and bond breaking involving sulfur) at relatively small computational cost. (I'm not suggesting the authors redo all their calculations, this is just a suggestion for future studies).

-Page 3, line 29: Most of the systems treated here were radicals, I assume with a spin multiplicity of 2. Did the authors use UM06-2X or ROM06-2X? (I assume the former, if so this could be stated, and also spin contamination values could be briefly discussed). Similarly, did the authors do UHF-UCCSD(T) or ROHF-ROCCSD(T)? Both can be done with Gaussian 09.

-Page 4, line 28 (also Page 6, line 1): the second reactions studied is not "barrierless" as such, there is a TS but it is far below the reactants. The proper term in this case would be a "submerged barrier". (The existence of a TS is also assumed by the use of equation 5). Note: the overall kinetic treatment seems fine, this is just an issue of terminology.

[Figure]

-Page 7, line 10: Please give some details about the "charge analysis" that was performed to determine that the oxygen molecules are formed in the singlet state. Also, could this not simply be an artefact of the computational method used? I assume the overall spin multiplicity is set to "2" - it would be very difficult to find structures corresponding to a radical plus a triplet oxygen (or two triplet oxygens!) in a DFT calculation on the doublet surface... Or in other words, the channels leading to the singlet oxygen molecules found by the authors are probably real, but there might also be channels (not discoverable with single-reference methods) leading to triplet oxygen. However as already the channels going to singlet oxygen are thermodynamically allowed, and kinetically fast, then this would not affect the conclusions. But the possibility could be stated. Also, please state clearly if BOTH of the formed $O_2$ molecules are expected to be in the singlet state.

-Page 7, line 23: It could be mentioned earlier on in the manuscript that the bonds between the $SO_2$ and $O_2$ moieties in "O2SOO-" are also co-ordination bonds, not proper covalent bonds. This makes it easier to understand how the conversion of O3...O2SOO- to O2...O2S-O3- can be barrierless (breaking a covalent bond would usually be associated with a barrier).

-Page 8, line 4: how can TSW2 be located below RCW2 that it connects to? Is this some entropy effect (i.e. the TS is higher in energy but lower in free energy)? Please explain.

-Page 8, line 9. A unimolecular rate of $6.5x1E15$ 1/s is unphysical, as it is faster than the typical frequency of molecular vibrations. This indicates that the used form of TST is not really applicable to this reaction where the TS is below the reactant (see above for a question on that). The conclusion that the reaction is extremely fast and likely occurs before any collisions with $N_2$ is valid, it's just the numerical value that doesn't make sense.

-Page 11 line 1: Why do the authors assume that OH and NOx are the main terminating

scavengers? Some other radicals, not to mention atmospheric acids, can easily have larger concentrations, and could thus increase the denominator of Eq 10.

-Page 11, line 9: Is it a good assumption that [O3] » [SO2]? The authors perform their calculations with [SO2] set to 5 ppb - [O3] certainly exceeds this in polluted areas, but not by many orders of magnitude, and in cleaner areas [O3] may not be much greater than this. . .

-Figure 2: I don't understand how RC1 can exist as a distinct minimum (stationary point) if there is no TS between it and P1. Or is this an energy/free energy issue, with RC1 below P1 in energy but above it in free energy? This should be discussed/mentioned - perhaps the potential energy surface could be shown also in terms of electronic energy, not just Gibbs free energy.

---

## Referee Comment (RC2) · Anonymous Referee #2 · 28 Nov 2018

Authors are Narcisse Tchinda Tsona and Lin Du

Recommendation: This paper is publishable subject to minor revisions noted. Further review is not needed.

Comments: The authors have theoretically studied the role of the superoxides such as O2- in SO2 oxidation. The simulations were performed using a combination of DFT and CC levels of theory (CCSD(T)/aug-cc-pVTZ//M06-2X/aug-cc-pVTZ). Calculations were performed in the gas phase with and without one water molecule. Two possible mechanisms for the titled reaction were suggested and considered. It is well known, that sulfur oxidation products play an important role in the atmosphere: formation of sec-

ondary aerosols, clouds and acid rains. Therefore, this theoretical work is an important contribution to a better understanding of the total mechanism of atmospheric sulfuric acid formation. I recommend publication this paper in the Atmospheric Chemistry and Physics Discussions after minor revisions.

Page 1, line 18. Misspelling the word "modelling". It does not need double LL. Page 3, line 9. Does not need a dot in the middle of sentences "with O3. in" Page 6, line 10. Delete the empty space between "4. 5" Page 12, line 16. Need to add a negative charge to the formula O2SOO.

1) In the Methods section, please, mention what multiplicity and charge did you use for the calculations of considered systems. 2) Did you perform IRC (intrinsic reaction coordinate) analysis, to prove that all your saddle points from the same PES (Pre-reactive complex – TS – Products)? If not, you should do it. 3) Do you think that just one water molecule is it a sufficient model to simulate liquid phase? Authors can additionally apply PCM models to the monohydrated system. Probably, in this case, the reaction will run spontaneously, without pre-reactive complex and TS (now, authors have a situation where in the case monohydrated system the energy of TSW2 is lower than the energy of the pre-reactive complex RCW2). 4) Please, add [Units] to the mentioned constants in Eq. 4 and 5 (for q, h, $\varepsilon 0$ etc.)

Please also note the supplement to this comment:
https://www.atmos-chem-phys-discuss.net/acp-2018-1111/acp-2018-1111-RC2-supplement.pdf

---

## Author Comment (AC1) · 20 Dec 2018

We thank the Referee for the constructive comments on our manuscript. Below are our point-to-point replies to the different questions raised by the Referee. For clarity, the Referee's comments are reproduced in blue color text and modified/inserted text in the revised manuscript are in red color text.

**Anonymous Referee #2**

Authors are Narcisse Tchinda Tsona and Lin Du

Recommendation: This paper is publishable subject to minor revisions noted. Further review is not needed.

Comments: The authors have theoretically studied the role of the superoxides such as O2- in SO2 oxidation. The simulations were performed using a combination of DFT and CC levels of theory (CCSD(T)/aug-cc-pVTZ//M06-2X/aug-cc-pVTZ). Calculations were performed in the gas phase with and without one water molecule. Two possible mechanisms for the titled reaction were suggested and considered.
It is well known, that sulfur oxidation products play an important role in the atmosphere: formation of secondary aerosols, clouds and acid rains. Therefore, this theoretical work is an important contribution to a better understanding of the total mechanism of atmospheric sulfuric acid formation. I recommend publication this paper in the Atmospheric Chemistry and Physics Discussions after minor revisions.

**Referee's comment:**
Page 1, line 18. Misspelling the word "modelling". It does not need double LL.

**Authors' reply**
This has been corrected.

**Referee's comment:**
Page 3, line 9. Does not need a dot in the middle of sentences "with O3. in"

**Authors' reply**
This has been corrected.

**Referee's comment:**
Page 6, line 10. Delete the empty space between "4. 5"

**Authors' reply**
This has been corrected.

**Referee's comment:**
Page 12, line 16. Need to add a negative charge to the formula O2SOO.

**Authors' reply**

This has been corrected.

**Referee's comment:**

1) In the Methods section, please, mention what multiplicity and charge did you use for the calculations of considered systems.

**Authors' reply**

The sentence at Page 3, Line 25-27 was re-written as:

"As the substrate in this study is a radical anion, all stationary points in the energy surface were optimized using density functional theory (DFT) based on the UM06-2X density functional (Zhao and Truhlar, 2008) and the aug-cc-pVTZ basis set (Dunning Jr et al., 2001), setting the charge to -1 and the spin multiplicity to 2."

**Referee's comment:**

2) Did you perform IRC (intrinsic reaction coordinate) analysis, to prove that all your saddle points from the same PES (Pre-reactive complex – TS – Products)? If not, you should do it.

**Authors' reply**

The IRC analysis was performed as indicated at Page 3, Line 4, they indeed connected the transition states to the pre-reactive complexes and the products.

**Referee's comment:**

3) Do you think that just one water molecule is it a sufficient model to simulate liquid phase? Authors can additionally apply PCM models to the monohydrated system. Probably, in this case, the reaction will run spontaneously, without pre-reactive complex and TS (now, authors have a situation where in the case monohydrated system the energy of TSW2 is lower than the energy of the pre-reactive complex RCW2).

**Authors' reply**

The reaction was performed in the gas-phase, exclusively. In the gas-phase, it was shown that only one water molecule can attach to O2SOO- and we also verified from our calculations that the addition of a second water vapor molecule to RC2 is not favorable under atmospheric conditions. It is, however, possible that O2SOO- would bind several water molecules in the liquid phase and the reaction properties would then be greatly affected.

The energy of TSW2 being lower than the energy of RCW2 is likely the effect of correlation since without CCSD(T) correction, TSW2 lies above RCW2. This difference in energy can also be explained by the difference in the electronic configurations of the two outer oxygen atoms of the $O_3$ moiety in the two transition states.

This information has been updated in the revised manuscript at Page 9, Lines 16-18 as:

"Another reason for this substantial drop in energy barrier is the difference in the electronic configurations of the two outer oxygen atoms of the $O_3$ moiety in the two transition states that form $O_2$ with different multiplicities in the products."

**Referee's comment:**
4) Please, add [Units] to the mentioned constants in Eq. 4 and 5 (for q, h, ε0 etc.)

**Authors' reply**

The units used in Eq. (4) are CGS units, whereas SI units are used in Eq. (5).

| Constants and variables | CGS units | SI units |
|---|---|---|
| $T$ | K | K |
| $k_B$ | $1.38 \times 10^{-16}$ erg K$^{-1}$ | $1.38 \times 10^{-23}$ J K$^{-1}$ |
| $h$ | $6.63 \times 10^{-27}$ erg s | $6.63 \times 10^{-34}$ J s |
| $q$ | $4.80 \times 10^{-10}$ statC | $1.60 \times 10^{-19}$ C |
| $\varepsilon_0$ | $1/(4\pi)$ | $8.85 \times 10^{-12}$ F m$^{-1}$ |
| $\mu$ | g | kg |
| $\alpha$ | cm$^3$ | F m$^2$ |
| $\alpha_D$ | StatC cm | C.m |

The following related sentences were added in the manuscript.
Page 6, Lines 13-15:
"The constants and variables in Eq. (4) and Eq. (5) are given in centimetre-gram-second (CGS) system of units and International System (SI) units, respectively. Details on these units are given in the Supplement."

---

## Author Comment (AC2) · 20 Dec 2018

We thank the Referee for the constructive comments on our manuscript. Below are our point-to-point replies to the different questions raised by the Referee. For clarity, the Referee's comments are reproduced in blue color text and modified/inserted text in the revised manuscript are in red color text.

**Anonymous Referee #1**

Tsona and Du have studied the reaction of O3 with O2SO2- using computational methods. This extends and complements a series of similar studies performed by the authors and their co-workers, aimed toward understanding the ionic contribution to sulfur chemistry in the atmosphere. The computational methods are broadly appropriate to the task (see below for two minor caveats on this), and the results are interesting for atmospheric chemists despite the studied SO2 oxidation pathway ultimately being rather minor compared to neutral channels. Overall, the text is understandable, though there are a large number of odd word choices and formulations - some proofreading or copy-editing would improve the manuscript. I recommend the manuscript be published in ACP subject so some minor revisions.

**Comments:**
-Page 2, Line 5-7: The authors list different oxidation routes for SO2 in the atmosphere: OH, sCIs, ions and mineral dust. The first three are gas-phase processes, the latter I assume corresponds to heterogeneous SO2 oxidation. However, there are many more heterogeneous pathways for SO2 oxidation, many of which are likely even more important than mineral dust, such as aqueous-phase oxidation inside cloud droplets. These could thus be mentioned.

**Authors' reply**
We sincerely thank the Referee for this reminder. The multiphase oxidation of $SO_2$ is an important path for sulfuric acid formation in the atmosphere and thus, this has been updated in the revised manuscript
The sentence at Page 2, Lines 3-5 has been modified as:
"Sulfur dioxide ($SO_2$), the most abundant sulfur-containing molecule in the atmosphere, is known to react both in the gas-phase and in multiphase oxidation processes following different mechanisms to form sulfate as the final oxidation species."
The following sentence has been inserted at Page 2, Lines 8-10:
"The main routes for $SO_2$ heterogeneous/multiphase oxidation include reactions with mineral dust (Harris et al., 2013), $O_3$ and $H_2O_2$ in cloud droplets (Caffrey et al., 2001; Hoyle et al., 2016; Harris et al., 2012; Hegg et al., 1996), $NO_2$ and $O_2$ in aerosol water and on $CaCO_3$ particles (Cheng et al., 2016; Wang et al., 2016; Zhang et al., 2018; Yu et al., 2018; Zhao et al., 2018)."

**Referee's comment:**
-Page 2, Line 11: The role of ions in aerosol formation has indeed been well established, and that role is in most conditions essentially "real but small". I.e. ionic pathways make a non-negligible contribution to aerosols, but in most atmospheric conditions the neutral pathways still dominate. This could be mentioned.

**Authors' reply**

It has been indeed demonstrated by previous studies that the contribution of ions in atmospheric particle formation is relatively small. To update this information in the revised manuscript, the text at Page 2, Lines 13-18 has been modified as follows:

"Sulfate is known to be the main driving species in atmospheric aerosols formation and its formation is critical in the determination of aerosol formation rates (Nieminen et al., 2009; Sipila et al., 2010; Kuang et al., 2008; Kulmala et al., 2000). The role of ions in this formation has been well established (Yu, 2006; Yu and Turco, 2000, 2001; Enghoff and Svensmark, 2008; Kirkby et al., 2011; Wagner et al., 2017; Yan et al., 2018), although relatively minor compared to the mechanism involving neutral particles, exclusively (Eisele et al., 2006; Manninen et al., 2010; Kirkby et al., 2011; Hirsikko et al., 2011; Wagner et al., 2017)"

**Referee's comment:**

-Page 3, Line 19: for sulfur-containing compounds, would it not be better/safer to use the aug-cc-pV(T+d)Z basis set instead of aug-cc-pVTZ? This should provide additional accuracy (especially for bond formation and bond breaking involving sulfur) at relatively small computational cost. (I'm not suggesting the authors redo all their calculations, this is just a suggestion for future studies).

**Authors' reply**

The treatment of the extra charge in anionic species is a known challenge to density functional theory in general. Extra electrons of anions are known to occupy diffuse, long ranging orbitals and, therefore, require special density functionals and basis sets for their accurate treatment. While for sulfur-containing species it is recommended when using the Dunning type basis sets to include extra *d* functions for the sulfur atom, Bork et al. recently showed that this is not desirable, for example, in predicting electron affinities, and they used the aug-cc-pVTZ basis set to study a reaction involving an electron transfer process (Bork et al., 2013). Using the CAM-B3LYP functional, they found that the aug-cc-pVTZ basis set gives a much better agreement with experiment than aug-cc-pV(T+d)Z, when calculating the difference between the electron affinities of $O_3$ and $SO_3$ (Bork et al., 2013). It should, however, be noted that the type of density functional used might also play a non-negligible role. Since our calculations involve similar species than in the above-mentioned study and also induce electron transfer, we used the aug-cc-pVTZ in our study.

**Referee's comment:**

-Page 3, line 29: Most of the systems treated here were radicals, I assume with a spin multiplicity of 2. Did the authors use UM06-2X or ROM06-2X? (I assume the former, if so this could be stated, and also spin contamination values could be briefly discussed). Similarly, did the authors do UHF-UCCSD(T) or ROHF-ROCCSD(T)? Both can be done with Gaussian 09.

**Authors' reply**

We used a spin multiplicity of 2 and hence the UM06-2X variant of the M06-2X density

functional and the UCCSD(T) variant of CCSD(T) were used in our calculations. This has been updated in the revised manuscript.

It is well-known that the use of UM06-2X as other unrestricted density functionals gives rise to spin contamination in the studied system. The spin contamination of all states was evaluated and found to be negligible for the electronic states in reaction (R2) while being important for the states in reaction (R3). The latter is likely due to $O_2$ formation in the pre-reactive complexes, transition states and products. In reaction (R3), the spin contamination is lowest is in the pre-reactive complexes and highest in the products, as a result of $O_2$ formation with different multiplicities. We added the following in the revised manuscript to account for spin contamination.

Page 3, Lines 27-32

"The use of UM06-2X implies using unrestricted wavefunctions to describe the quantum state of the system. Spin contamination often arises from unrestricted density functional theory (DFT) calculations and it is not guaranteed that the electronic states from these calculations are eigenstates of the $\hat{S}^2$ operator. The spin contamination was then evaluated for all electronic states as $\Delta S = \langle \hat{S}^2 \rangle - \langle \hat{S}^2 \rangle_{ideal}$, where $\langle \hat{S}^2 \rangle$ is the actual value of the expectation value of the $\hat{S}^2$ operator from DFT calculations and $\langle \hat{S}^2 \rangle_{ideal}$ is the ideal expectation value. For systems explored in this study, $\langle \hat{S}^2 \rangle_{ideal} = 0.75$."

Page 6, Line 30

"The spin contaminations for RC1 and RCW1 are negligible, being 0.0086 and 0.0081, respectively."

Page 9, Lines 27-32

"The spin contamination for electronic states in reaction (R3) is quite significant, being 1.0122, 1.4666, and 2.0374 for the pre-reactive complex, transition state and product, respectively, and is almost insensitive to the presence of water. The actual values of the expectation values of the $\hat{S}^2$ operator for all electronic states obtained from our calculations are given in the Supplement, along with their cartesian coordinates. The high values of spin contamination likely reflect the formation of $O_2$ with different multiplicities within the system. As the charge analysis indicates, starting with singlet $O_2$ in the pre-reactive complexes of reaction (R3), both singlet and triplet $O_2$ are formed in the final products."

**Referee's comment:**

-Page 4, line 28 (also Page 6, line 1): the second reactions studied is not "barrierless" as such, there is a TS but it is far below the reactants. The proper term in this case would be a "submerged barrier". (The existence of a TS is also assumed by the use of equation 5). Note: the overall kinetic treatment seems fine, this is just an issue of terminology.

**Authors' reply**

The word "barrierless" at Page 4 and Page 6 has been changed to "submerged barrier" and "low-lying" and the new sentences in the revised manuscript now read as:

Page 5, Line 17

"This consideration is, however, not valid for reactions with submerged barrier, since the prereactive intermediate seldom thermally equilibrates."

Page 6, Line 22

"The second process is the low-lying formation of a molecular complex in which the $SO_2$ entity of $O_2SOO^-\cdots(H_2O)_{0-1}$ is oxidized to $SO_3^-$."

**Referee's comment:**

-Page 7, line 10: Please give some details about the "charge analysis" that was performed to determine that the oxygen molecules are formed in the singlet state. Also, could this not simply be an artefact of the computational method used? I assume the overall spin multiplicity is set to "2" - it would be very difficult to find structures corresponding to a radical plus a triplet oxygen (or two triplet oxygens!) in a DFT calculation on the doublet surface... Or in other words, the channels leading to the singlet oxygen molecules found by the authors are probably real, but there might also be channels (not discoverable with single-reference methods) leading to triplet oxygen. However as already the channels going to singlet oxygen are thermodynamically allowed, and kinetically fast, then this would not affect the conclusions. But the possibility could be stated. Also, please state clearly if BOTH of the formed O2 molecules are expected to be in the singlet state.

**Authors' reply**

The electronic charge analysis, according to the Bader charge partitioning, is an intuitive way of dividing molecules into atoms, which are purely defined based on electronic charge density (Bader, 1998). This approach assumes that the charge density in molecular systems reaches a minimum between atoms, and this minimum density is a natural place to separate atoms from each other (Bader, 1998; Henkelman et al., 2006). The calculations performed by this method produce different output files among which, a file that can be visualized and a file containing the electronic charge associated to each atom according to the Bader partitioning. In our calculations, these files allow to see how the extra electron (charge) is distributed over the atoms of the system. Due to the possibility of the $O_2$ molecule to form both in its singlet and triplet states, the Bader charge partitioning can equally indicate the presence of free electrons, especially in the case of the triplet.

For Reaction (R3), the files that can be visualized were plotted for the unhydrated system and given in **Fig. 3** in the main manuscript, wherefrom the electron cloud on atoms can be seen. Examining the files containing the electronic charge associated to each atom, it is seen that the electronic charge on the pre-reactive complex is essentially on two oxygen atoms of the $O_3$ moiety that is coordinated to $SO_2$. The net charge on these atoms is 1.04e, whereas the net charge on the free $O_2$ molecule is 0.01e. The latter value shows that the $O_2$ molecule released in the optimization of the pre-reactive complex (**Fig. 3(a)**) has no unpaired electrons, indicating that this $O_2$ molecule is in its singlet state.

In the transition state structure (**Fig. 3(b)**), the charge is still on the $O_3$ moiety, although mostly located on the oxygen atom bound to sulfur. The net charge on the two outer oxygen atoms of $O_3$ that will form the $O_2$ molecule in the product state has substantially decreased to 0.30e while the charge on the free $O_2$ molecule has slightly increased to 0.04e. The free $O_2$ molecule can

[revised manuscript text omitted]

**Referee's comment:**

-Page 7, line 23: It could be mentioned earlier on in the manuscript that the bonds between the SO2 and O2 moieties in "O2SOO-" are also co-ordination bonds, not proper covalent bonds. This makes it easier to understand how the conversion of O3…O2SOO- to O2…O2S-O3- can be barrierless (breaking a covalent bond would usually be associated with a barrier).

**Authors' reply**

Indeed, the $O_2S–OO^-$ bond is a coordination bond rather than covalent and this has been clarified in the revised manuscript by re-writing the sentence at Page 2, Lines 30-32 as:

"A previous study demonstrated that two forms of $SO_4^-$ separated by a high energy barrier may exist in the atmosphere (Tsona et al., 2014): the sulfate radical ion henceforth indicated as $SO_4^-$, and the peroxy form, $O_2SOO^-$, in which the $O_2S–OO^-$ bond nature is more dative than covalent."

**Referee's comment:**

-Page 8, line 4: how can TSW2 be located below RCW2 that it connects to? Is this some entropy effect (i.e. the TS is higher in energy but lower in free energy)? Please explain.

**Authors' reply**

According to our calculations, the density functional theory (DFT) calculations based on the UM062X/aug-cc-pVTZ method predict the TSW2 configuration to be located ~9 kcal mol$^{-1}$ electronic energy above RCW2. However, upon correction by the UCCSD(T)/aug-cc-pVTZ method the energy of the transition state considerably drops to 3 kcal mol$^{-1}$ below that of the RCW2. As an uncommon fact for atmospheric reactions and since this could not be assigned to entropic effects, we first speculated that wrong structures for RCW2 and TSW2 would have been optimized. We then repeated the calculations twice and found that the results were similar to the previous case. For each trial, although the UM062X/aug-cc-pVTZ calculations gave TSW2 located above RCW2, the UCCSD(T)/aug-cc-pVTZ correction reversed the situation. Most likely, the low-lying TSW2 is due the correlation effect on the electronic energy since without UCCSD(T) correction, TSW2 lies above RCW2. This can further be explained by the difference in the electronic configurations of the two outer oxygen atoms of the $O_3$ moiety in the TS2 and TSW2 transition states, as clarified in our reply to a comment above. Further clarification is given in the manuscript at Page 9, Line 16-18 as:

"Another reason for this substantial drop in energy barrier is the difference in the electronic

configurations of the two outer oxygen atoms of the $O_3$ moiety in the two transition states that form $O_2$ with different multiplicities in the products."

**Referee's comment:**
-Page 8, line 9. A unimolecular rate of 6.5x1E15 1/s is unphysical, as it is faster than the typical frequency of molecular vibrations. This indicates that the used form of TST is not really applicable to this reaction where the TS is below the reactant (see above for a question on that). The conclusion that the reaction is extremely fast and likely occurs before any collisions with N2 is valid, it's just the numerical value that doesn't make sense.

**Authors' reply**
With the negative energy barrier of reaction (R3) in the presence of water, the overall rate constant is essentially collision limited as indicated by the reported value of $8.0\times10^{-10}$ $cm^3$ $molecule^{-1}$ $s^{-1}$. As explained in the manuscript, the outer transition state provides the dominant bottleneck to the rate constant of reaction (R3) in the presence of water, which can then be calculated directly by eq. (4). To avoid using unphysical number, the unimolecular rate constant value of $6.5\times10^{15}$ $s^{-1}$ is deleted in the revised manuscript and the sentence at page 9, Lines 20-23 is revised as:

"Based on the TS2 energy, the unimolecular decomposition of $O_2\cdots O_2S–O_3^-$ at 298 K in the absence of water was found to occur at a rate constant of $3.1\times10^5$ $s^{-1}$, corresponding to an atmospheric lifetime of 3.3 µs. Both this short lifetime and the negative energy barrier of the monohydrated reaction indicate that $O_2\cdots O_2S–O_3^-$ would not live long enough to experience collisions with other atmospheric oxidants."

**Referee's comment:**
-Page 11 line 1: Why do the authors assume that OH and NOx are the main terminating scavengers? Some other radicals, not to mention atmospheric acids, can easily have larger concentrations, and could thus increase the denominator of Eq 10.

**Authors' reply**
Though we initially focused on OH and NOx as scavengers that may form well-known stable $HSO_4^-$ and $NO_3^-$ species, it is reasonable that other species like $HO_2$ and organic acids which have relatively high concentrations may be good scavengers as well. They may form $HSO_4^-$ and $CO_3^-$ to scavenge the free electron. Noting that organic acids are mainly dominated by formic and acetic acid in the atmosphere, their concentrations are considered in Eq. (9) as representative examples of organic acids. Taking into account $10^8$ molecule $cm^{-3}$ concentration for $HO_2$ (Holland et al., 2003; Dusanter et al., 2009) and 110 ppt = $2.4 \times 10^9$ molecule $cm^{-3}$ for organic acids (formic acid and acetic acid) (Le Breton et al., 2012; Baasandorj et al., 2015) in Eq. (9), the contribution of ion-induced $SO_2$ oxidation to $H_2SO_4$ formation is in the 0.1-2.0% range depending on the altitude. This has been updated in the revised manuscript.
The sentence at Page 12, Line 14 was modified as:
"The main ones are likely NOx, OH, $HO_2$ and organic acids, which lead to the formation of the stable $NO_3^-$, $HSO_4^-$, and $CO_3^-$ species."

The text at Page 12, Line 33 to Page 13, Lines 1-5 was modified as:

"We assume nearly pristine conditions with [$SO_2$] = 5 ppb = $1.2 \times 10^{11}$ molecule cm$^{-3}$, [NOx] = 200 ppt = $4.9 \times 10^9$ molecule cm$^{-3}$, [OH] = $5.0 \times 10^5$ molecule cm$^{-3}$ (day and night average), and [$HO_2$] = $10^8$ molecule cm$^{-3}$ (Dusanter et al., 2009; Holland et al., 2003). Noting that formic acid and acetic acid are the most abundant organic acids in the atmosphere, their concentrations are considered in Eq. (9) as representative examples for organic acids, [organic acids] = 110 ppt = $2.7 \times 10^9$ molecule cm$^{-3}$ ( Le Breton et al., 2012; Baasandorj et al., 2015). We then determine $J_{ion}$ in the range $3.2 \times 10^1$–$1.6 \times 10^3$ cm$^{-3}$ s$^{-1}$."

The sentence at Page 13, Line 14 was modified as:

"We find that the contribution of ion-induced $SO_2$ oxidation to $H_2SO_4$ formation can range from 0.1 to 2.0% of the total formation rate."

Equation (9) has been modified to

$$TON \approx \frac{[SO_2]}{[OH]+[HO_2]+[NO_x]+[organic\ acids]} \qquad (9)$$

**Referee's comment:**
-Page 11, line 9: Is it a good assumption that [O3] » [SO2]? The authors perform their calculations with [SO2] set to 5 ppb - [O3] certainly exceeds this in polluted areas, but not by many orders of magnitude, and in cleaner areas [O3] may not be much greater than this…

**Authors' reply**
Although the [$O_3$] » [$SO_2$] assumption might be somewhat overestimated, the fact remains that [$O_3$] > [$SO_2$] as indicated also by the referee. This condition reinforces the condition that $SO_2$ should be the limiting factor in the reaction process, which was our original idea. Hence, to remove the confusion concerning the $O_3$ concentration relative to that of $SO_2$, we simply changed [$O_3$] » [$SO_2$] to [$O_3$] > [$SO_2$] in the revised manuscript and re-wrote the sentence at Page 12, Lines 22-23 as:

"The concentration of the catalyst can be approximated to the concentration of the scavengers and, considering that at most atmospheric conditions [$O_3$]>[$SO_2$], $SO_2$ is the limiting species in reaction (R4)."

**Referee's comment:**
-Figure 2: I don't understand how RC1 can exist as a distinct minimum (stationary point) if there is no TS between it and P1. Or is this an energy/free energy issue, with RC1 below P1 in energy but above it in free energy? This should be discussed/mentioned - perhaps the potential energy surface could be shown also in terms of electronic energy, not just Gibbs free energy.

**Authors' reply**
Reaction (R2) involves a charge transfer, followed by cluster decomposition. The immediate outcome of $O_2SOO^- + O_3$ optimization is RC1, formed with -5.1 and 4.5 kcal mol$^{-1}$ electronic and Gibbs free energy, respectively. While the electronic energy, which defines the best

possible arrangement of the electrons in the system, shows that the formation of RC1 is possible at 0 K, the positive Gibbs free energy at 1 atm and 298 K indicates that this complex is not stable under these atmospheric conditions and will surely decompose or react further upon formation. The RC1 formation represents the lowest state at which $O_2SOO^-$ can interact with $O_3$ to allow electron transfer and $O_2S–OO$ decomposition. The energy of this state then corresponds to the energy barrier to form the $O_2 + SO_2 + O_3^-$. For more clarification, the following texts were added in the revised manuscript

The sentence at Page 6, Line 31 to Page 7, Lines 1-3 was modified as:

"The electronic energies of formation of RC1 and RCW1 are -5.1 and -4.6 kcal mol$^{-1}$, respectively. Despite these complexes may form at 0 K, the Gibbs free energies of their formation under atmospheric pressure and 298 K (4.5 and 4.7 kcal mol$^{-1}$, respectively) indicate that their formation is endergonic under atmospherically relevant conditions."

The following was inserted at Page 7, Lines 4-6:

"Hence, the Gibbs free energies of formation of RC1 and RCW1 define the lowest states at which $O_2SOO^-$ can interact with $O_3$ to allow electron transfer and $O_2S–OO$ decomposition, and thus represent the energy barrier towards $O_2 + SO_2 + O_3^-$ formation."